# Supraclavicular versus infraclavicular approach in inserting totally implantable central venous access for cancer therapy: A comparative retrospective study

Amine Souadka[1]*, Hajar Essangri[1], Imad Boualaoui[1], Abdelilah Ghannam[2], Amine Benkabbou[1], Laila Amrani[1], Raouf Mohsine[1], Mohammed Anass Majbar[1]

1 Surgical Oncology Department, National Institute of Oncology, Mohammed V University Medical School, Rabat, Morocco, 2 Anesthesia and Intensive Care Department, National Institute of Oncology, Mohammed V University Medical School, Rabat, Morocco

* a.souadka@um5s.net.ma

## Abstract

### Introduction

The insertion of an implantable central venous access is performed according to a variety of approaches which allow the access to the subclavian vein, yet the supraclavicular technique has been underused and never compared to the other methods. The aim of this study was to testify on the efficacy and safety of the subclavian puncture without ultrasound guidance « Yoffa » in comparison with the classical infraclavicular approach (ICA).

### Material and methods

This is a retrospective study with prospective data collection on patients followed at the national oncology institute for cancer, in the period extending from May 1st 2017 to August 31st 2017. All patients had a totally implantable central venous access device inserted by the same surgeon AS for chemotherapy administration and demographic characteristics, as well as procedure details were examined. The primary outcomes were the intraoperative complications, while the secondary outcomes represented immediate postoperative and mid-term complications (at 15 months of follow up). Outcomes were compared between techniques by means of non parametric tests and the Fischer test.

### Results

Our study included 135 patients with 70 patients undergoing the subclavian technique, while 65 were subject to the infraclavicular approach. Both groups had no statistically significant demographic characteristics. The number of vein puncture attempts exceeding once, the accidental artery puncture and operative time were more significant in the ICA group; (39,6 vs 17,6 p = 0,01) (9.2% vs 0; p = 0,01) and (27± 13 vs 23± 8min, p = 0.045) respectively. There was no statistically significant difference in the immediate and midterm complication rate between the two methods 1(1,4) vs 2 (3) p = 0.5.

**Data Availability Statement:** All relevant data are within the paper and its Supporting Information files.

**Funding:** This study received finantial support from l'Institut de Recherche sur le Cancer (IRC (Morocco) for the payment of the APC of this journal The funders had no role in study design, data collection and analysis, decision to publish, or preparation of the manuscript. The authors received no specific funding for this work.

**Competing interests:** The authors have declared that no competing interests exist.

## Conclusion

In case of unavailability of ultrasonographic guidance, the use of the supra-clavicular landmarks approach is linked to higher success rates and less arterial punctures, thereby proving to be a safe and reliable approach.

## Introduction

The first placement of a totally implantable central venous access device (**TIVAD**) was performed in 1982 using the cephalic vein and ever since, TIVADs are standardly used in the oncology field for the administration of chemotherapeutic agents, intravenous hydration, parenteral nutrition and blood withdrawal. This tool has proven to provide constant and reliable parenteral access and reduce to some level the vascular complications of cancer chemotherapy, as well as improve the overall quality of life of these patients [1]. Yet many aspects of TIVADs use are still a matter of discussion, namely the choice of device, vein and technique of insertion.

The most frequently used central venous catheterization approaches vary between the internal jugular used for resuscitation and the subclavian vein which is often adopted during surgical procedures. The access to the subclavian vein can either be possible through the infra-clavicular or the supra-clavicular approach, with the latter being less frequently used. This supraclavicular technique for central vein puncture, first described In 1965 by Yoffa [2], was mainly used in emergency resuscitation by anaesthesiologist as a quicker way to central line catheterization, allowing higher first attempt success rate and accessibility during CPR, tube thoracostomy and active surgery [3, 4]. In addition, the lack of resources such as ultrasonographic (**US**) guidance—which has been standardized in TIVAD and central line access insertion—has put the blind approach relying on landmark localization for vein puncture on the front line as well as prompted investigating these methods [5–7].

We wanted to examine the potential benefits of the supraclavicular puncture in terms of efficacy, safety and intra interventional pain perception. The aim of this study is to compare the supraclavicular puncture to the classical infra-clavicular approach, both without ultrasound guidance and demonstrate the efficacy and safety of this underused technique.

## Methods

### Patients

This is a retrospective analysis of a prospective database conducted at the National Institute of Oncology (**NIO**) of Rabat, Morocco in the period from May 1st 2017 to August 31st 2017. All patients with the indication of systemic chemotherapy and TIVAD insertion by a single senior surgeon (A.S) were eligible and admitted in a day-hospital setting. Only Patients meeting the following criteria were included: patients aged > 18 year old, with an Eastern Cooperative Oncology Group (**ECOG**) performance status of 0 to 2, able to bear a lying position for at least 45 minutes and diagnosed with mainly digestive or gynecologic cancer.

Patients with superior vena cava syndrome, active infection, coagulopathy (defined as a platelet count <50 000/l and/or a prothrombin rate < 50%), life expectancy <6 months or the inability to sign the written informed consent were excluded.

## Implantable ports and access to central vein

Patients underwent the insertion of a single type of silastic port with a silicone membrane and connected to an 8F silicone rubber catheter (Bard Port, Bard Inc., Salt Lake City, UT) using a percutaneous landmark access approach to the subclavian vein by a single surgeon skilled in both techniques. Both approaches are standard procedures in our institution.

Patients were divided in 2 groups, depending on the TIVAD implantation technique and period in which they underwent the procedure. Group A patients received TIVAD through a supraclavicular approach and group B through an infraclavicular approach in the first and second half of our study period respectively (Table 1). The right side was attempted whenever possible. At the 3rd failed attempt, the surgeon would try the other puncture site and in case this second route was also unsuccessful, patients were excluded from the analysis.

## Technique

**Supraclavicular approach.**   Patients are placed in a Trendelenburg's position with slight shoulders extension and the procedure is performed under local anesthesia (20 cc of lidocaine 1%), following the previously described technique by David YOFFA [2].

The key success factor is the correct identification of the clavisternomastoid angle landmark formed by the junction of the lateral head of the sternocleidomastoid muscle and the clavicle using active rising of the patient's head.

The site of needle insertion is 0.5 to 1cm lateral to the clavicular head of the sternocleido-mastoid muscle and 1 cm posterior to the clavicle. The needle is oriented at a 45˚ angle to the sagittal and transverse planes and a 15˚ angle below the coronal plane with the contralateral nipple as a target for directing the introducer [8]. Subsequently to the vein puncture, a guide-wire is inserted through the needle which is withdrawn and replaced by a 'peel-away sheath' introducer the time of the completion of the second step of the procedure by the surgeon.

**Table 1. Demographics of patients.**

| | Group A (YOFFA) | Group B (ICA) | P value |
|---|---|---|---|
| | N = 70 | N = 65 | |
| Mean age (years) ± DS | 54 ± 16 | 51 ± 16 | 0.427 |
| Gender N (%) | | | 0.9 |
| • Female | 34 (38.6) | 31 (47.7) | |
| • Male | 36 (51.4) | 34 (52.3) | |
| ASA | | | 0,88 |
| 1 | 39 (55) | 37 (57) | |
| 2–3 | 31 (45) | 28 (43) | |
| Mean BMI (Kg/m2) | 26 ± 3,2 | 24,6 ± 2,7 | 0,67 |
| Side of puncture, N (%) | | | 0.512 |
| • Right | 60 (86) | 53 (81) | |
| • Left | 10 (14) | 12 (19) | |
| Neoplastic Disease N (%) | | | 0.56 |
| • Digestive | 45 (64.3) | 40 (61.5) | |
| • Gynecologic | 18 (25.7) | 21 (32.3) | |
| • Others | 7 (10) | 4 (4.2) | |
| Chemotherapy administration before TIVAD insertion, N (%) | | | 0,06 |
| Yes | 24 (34,2) | 13 (20) | |
| No | 46 (65,8) | 52 (80) | |

**Infraclavicular approach.** Following the same positioning and initial steps, the landmark of needle insertion for this approach is 1 cm lateral to the middle third of the clavicle with a 10 to 15˚ incline from the coronal plane to the direction of the sternal notch or the contralateral acromioclavicular articulation.

In order to decrease malposition risk of the guidewire in the ipsilateral Internal jugular vein, a sterile finger may be placed in the corresponding supraclavicular fossa.

## Port, catheter insertion and control

A surgically dissected space is prepared for port placement, while being connected to the catheter.

A first surgical time (t1) is marked by skin incision and insertion of a tunneller from the dissected space to the venipuncture site to ensure a subcutaneous path for the catheter, as well as a pocket for the venous access device.

The second surgical time (t2), is marked by the advancement of the 'peel-away sheath' dilator into the vein. Following the removal of the dilator, the catheter is inserted through the vein and the port is then flushed by 20cc of saline after checking for blood reflux.

A chest X-ray was routinely performed following the catheter placement in order to visualize the location at the lower or middle part of the superior vena cava and rule out a pneumothorax. Antibiotic prophylaxis was not routinely administered.

## Endpoints

Primary outcomes were the intraoperative result defined by a rate of successful first attempt, the number of accidental artery puncture, the occurrence of an intraoperative incident that led to change in technique (ex: local Hematoma), difficulties identifying the landmarks as well as the total operative time and pain perception. During the procedure each patient was asked to individually rank the intra-interventional pain perception on a visual analogue scale (1 = no pain and 10 = maximum imaginable pain) at (t1) and (t2).

Secondary outcomes were surgery related complications either categorized as early complications, defined by the occurrence of infection, a hematoma or pneumothorax in the first week following the procedure and prior to the first use of the device, or late complication after assessment in March 2018 such as infection, thrombotic obstruction, fracture embolism (Fat emboli) or patient death leading to the non-use or removal of the device.

## Statistical evaluation

Continuous variables were presented with a mean value ± SD or a median with interquartile range and categorical variables were expressed with frequencies and percentages. For the pain perception evaluation, a Wilcoxon t-test was used. Categorical variables and the comparison of the two groups was performed using the $\chi2$ test or Fisher's exact test when appropriate. A p value of $<0.05$ was considered statistically significant. All statistical analysis was performed using SPSS software (SPSS 13.0; SPSS Inc, Chicago, IL).

## Results and discussion

### Results

During this study period, one hundred and thirty five patients underwent the successful placement of a TIVAD prior to chemotherapy with seventy (51.9%) patients in the group A (using the supraclavicular approach) versus sixty-five (48.1%) patients in the group B (using the infraclavicular approach). There were no significant differences between the two groups concerning

mean age, gender, American Society of Anesthesiologists (ASA) score, Body mass index (BMI), neoplastic disease and the side of puncture. Further demographic characteristics are shown in Table 1.

Regarding intraoperative outcomes, the first attempt success rate was 82.4% (56/70) vs. 61% (39/65) for group A and B respectively(p = 0.01). Group A had a null rate of accidental artery punctures compared to group B 6/65 (9.2%) with p = 0.01. The procedure was significantly longer in duration for group B (27± 13 vs 23± 8min, p = 0.045) and the surgeon had to switch the techniques in 9 cases vs 2 cases in group B and A respectively. This is mainly due to a failure at the 3$^{rd}$ attempt in 5/65 of cases from group B compared to 1/70 cases in group A. Table 2.

As regards pain perception, patients who chose the supraclavicular approach (groupe A) reported a more significant pain sensation at the time of the tunneller insertion from the dissected space to the venipuncture site (t1). Opposingly, the same group described less pain during the insertion of the dilator over the guidewire and into the vein.

No TIVAD insertion related death was observed, thus all patients completed at least 15 months of follow up.

Table 3 is a representation of both early and late postoperative outcomes for the two techniques. Only one early infection occurred in group A and one device defunctioning in group B, which required the interruption of the TIVAD use during the antibiotic treatment period for the first case and the device location change to the contralateral side in the other. No case of thrombosis was described after the 15 months follow up period, however, a pinch off syndrome was observed in group B leading to a late change to the second technique.

## Discussion

This study showed that the supraclavicular approach to TIVAD implantation is more successful (97.2% vs 86.2), shorter in duration (23 vs 27 min) and has a higher rate of success at first attempt with less arterial puncture incidents than the infraclavicular approach.

This approach offers many advantages compared to the infraclavicular puncture technique due to its easily localizable landmarks even in obese patients, the shorter needle insertion depth (smaller skin vein distance), the larger target area, the straighter route to the vena cava and reduced risk of lung, pleural or arterial injuries [9–12].

This explains the significantly higher success at first attempt rate of 86% and the global success rate of 97.2%, alongside the very few early and late complications. Only three comparative studies inspected the supraclavicular and infraclavicular approaches, with success rates ranging

**Table 2. Outcomes comparison between the Yoffa and the ICA technique groups.**

|  | Group A (Yoffa) | Group B (ICA) | P |
|---|---|---|---|
| Success at first attempt, N (%) | 56 (82.4) | 39 (61) | 0.01 |
| Accidental artery puncture, N (%) | 0 (0) | 6 (9.2) | 0.01 |
| Mean operative time (±SD) | 23±8 | 27± 13 | 0.042 |
| Success rate, N (%) | 68 (97.25) | 56 (86.2) | 0.019 |
| Cause of changing site of puncture, |  |  |  |
| N (%) | 2 (2.85) | 9 (13.8) | 0.019 |
| • Hematoma | 1 | 4 |  |
| • Failure at the 3rd attempt | 1 | 5 |  |
| Median Pain perception rate in scale from 1 to 10 (quartiles) |  |  |  |
| At (T1) | 1 (0–3) | 0 (0–2) | 0.001 |
| At (T2) | 1 (1–4) | 3 (1–5) | 0.001 |

**Table 3. Early and late complications (follow up 15 months) of surgical insertion of TIVAD.**

|  | Groupe A | Groupe B |
|---|---|---|
| Early complications |  |  |
| Infection | 1 (1.4) | 0 |
| dysfonction | 0 | 1(1.5) |
| Late complications |  |  |
| Thrombosis | 0 | 0 |
| Pinch Off syndrom | 0 | 1 (1.5) |

from 84.5% to 93.3% and 80% to 87% for both approaches respectively. The rates of complications were heterogeneous depending on the definitions [3, 13, 14].

Very few data in the literature addresses patient pain perspective in this type of procedure with only one study demonstrating the jugular catheter placement to be significantly less painful than the infra-clavicular insertion using the subjective pain scale [15]. Our study categorized pain assessment according to the two different painful moment of the procedure: **t1** being the time of tunneller insertion ensuring the subcutaneous path of the catheter, which was not significantly different in the two groups despite the longer distance from the dissected space to the vein puncture in the supraclavicular approach. Furthermore, the second surgical time **t2,** corresponding to the 'peel-away sheath' dilator insertion was significantly more painful in the subclavian approach which could probably be due to the proximity of the pleura and lung, as well as the length of the distance from the skin to the vein [9].

Although the use of ultrasound guidance (**US**) represents a significant advance in central line placement, following this approach in the subclavian vein puncture often includes some additional risks [16]. Although US guided procedures could increase the anatomical perception of the performers [17, 18], the false sense of security, routinely reliance on this tool and the complication risk resulting from the effect on healthcare professionals' landmark-based techniques could also be argued [19]. For this reason, landmark based approaches remain a skill physicians need to have in their armamentarium, which applies to TIVAD insertion as well.

Limitations of this study are the retrospective aspect, absence of randomization and the fact that the results of this study were directly dependent on the senior surgeon's experience and skill. Therefore, caution is required when interpreting and applying our results to procedures where novices or physicians, especially at the beginning of their learning curve, are attempting to perform successfully in both approaches. An additional limitation is the relatively small number of patients included, however, this will allow us to decide the required sample size for future studies addressing this technique.

## Conclusions

Nowadays, central venous access is performed under ultrasound guidance, which is not always possible in resource-restricted environments. In this case, the supraclavicular line is an attractive approach that appears to be safe and possibly easier to perform, with easier localizable landmarks and less misplacement than more frequently used lines and. As such, the supraclavicular approach could challenge more traditional techniques and more studies investigating this alternative are required.

## Supporting information

**S1 Data.**
(SAV)

## Acknowledgments

The authors would like to thank Miss Hanane Benkhouya for her support.

## Author Contributions

**Conceptualization:** Amine Souadka, Amine Benkabbou, Mohammed Anass Majbar.

**Data curation:** Amine Souadka, Imad Boualaoui.

**Formal analysis:** Amine Souadka.

**Methodology:** Amine Souadka.

**Supervision:** Abdelilah Ghannam, Raouf Mohsine.

**Validation:** Amine Souadka.

**Writing – original draft:** Amine Souadka, Hajar Essangri, Mohammed Anass Majbar.

**Writing – review & editing:** Amine Souadka, Hajar Essangri, Abdelilah Ghannam, Amine Benkabbou, Laila Amrani, Raouf Mohsine, Mohammed Anass Majbar.

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
