## [Decision Letter · Decision Letter 0]

28 Aug 2020

PONE-D-20-14056

Supraclavicular versus infraclavicular approach in inserting totally implantable central venous access for cancer therapy : a comparative study on safety, efficacy and intra interventional pain perception

PLOS ONE

Dear Dr. Souadka,

Thank you for submitting your manuscript to PLOS ONE. After careful consideration, we feel that it has merit but does not fully meet PLOS ONE’s publication criteria as it currently stands. Therefore, we invite you to submit a revised version of the manuscript that addresses the points raised during the review process.

Please address the reviewers' concerns and revise the manuscriopt accordingly.

We look forward to receiving your revised manuscript.

Kind regards,

Academic Editor

PLOS ONE

Journal Requirements:

2.In your Data Availability statement, you have not specified where the minimal data set underlying the results described in your manuscript can be found. PLOS defines a study's minimal data set as the underlying data used to reach the conclusions drawn in the manuscript and any additional data required to replicate the reported study findings in their entirety. All PLOS journals require that the minimal data set be made fully available. For more information about our data policy, please see http://journals.plos.org/plosone/s/data-availability.

3.Thank you for stating the following financial disclosure:

 [No].

4.Thank you for stating the following in your Competing Interests section: 

[No].

Reviewers' comments:

Reviewer's Responses to Questions

**Comments to the Author**

1. Is the manuscript technically sound, and do the data support the conclusions?

Reviewer #1: Yes

Reviewer #2: Yes

2. Has the statistical analysis been performed appropriately and rigorously? 

Reviewer #1: Yes

Reviewer #2: Yes

3. Have the authors made all data underlying the findings in their manuscript fully available?

Reviewer #1: No

Reviewer #2: No

4. Is the manuscript presented in an intelligible fashion and written in standard English?

Reviewer #1: Yes

Reviewer #2: Yes

5. Review Comments to the Author

Reviewer #1: This is a study comparing the supraclavicular and infraclavicular approaches for central venous access and for placing implantable venous access device. Both approaches were applied using landmark technique (without ultrasound guidance).

The main objective was to compare these two techniques in terms of procedural success and performance time.

The authors found a better first pass success rate and shorter procedure time for supraclavicular approach compared to infraclavicular method.

Plase find below my comments:

Major strengths of the study:

Comparison of one popular central access technique to a disregarded technique which may be a reliable alternative for placing TIVAD.

Major weaknesses of the study and comments for authors:

Ultrasound guidance is a standardized method for central venous access as the authors mentioned. In major institutions where TIVADs are placed, ultrasound guidance is essential for those procedures.

The patients receiving the two procedures were not randomized which may lead to selection bias (as this was a retrospective cohort study).

-Please mention type of the study method in the title.

Introduction:

The hypothesis and objective of the study should be explained in detail in the introduction part.

Methods:

-The sample size may not be sufficient to determine rarely seen early and late complications.

-In the sentence: “Group A patients received TIVAD through a supraclavicular approach and group B through an infraclavicular approach in the first and second half of our study period respectively (see details below).” Which details below where? A table? Insertion techniques?

Results

-Duration of catheter placement was mentioned in discussion part but there is no data in the results.

Discussion

-Personally, I do not agree with the authors about the presumption that ultrasound guided catheter placement may lead to deskilling. Although this is arguable, I believe that us guided procedures increase anatomicaly perception of the performers regarding the anatomic structures as they see them every time they perform the procedure (see anatomic variations as well).

I would like to declare that I have no conflict of interest.

Reviewer #2: Thank you for the chance to read your paper - it has highlighted a technique for access of which I was not previously aware.

The key limitations are already described by yourselves, and it is a shame as they would be 'easily' addressed by a longer study period and true randomisation rather than by time period.

Major comments:

Abstract:

1. "yet the supraclavicular technique has been underused and never compared to the other method".

No true - you yourselves reference several published comparisons!

2. (1,4% vs 7% p=0,21). This is a large difference without significance. Ranges should be included with the data to understand why this is so. In the results table the figure is 1.4 vs 1.5. Typo?

Introduction:

The text needs clarity. The techniques you are talking about need to be made very clear in the introduction, as at the moment it is a little confusing for me as a reader.

For example: Be clear in the abstract and the introduction that you are just assessing access to the subclavian vein.

"Accordingly, the most frequent central venous catheterization approaches vary

between the subclavian which is often adopted during surgical procedures and the

internal jugular used for resuscitation. The use of the supraclavicular approach on the

other hand is less usual. In fact, the supraclavicular technique for central vein puncture

was first described In 1965 by Yoffa [2]."

The phrasing of this paragraph makes it seem like you are discussing subclavian access to supraclavicular IJV access, due to structure. If unfamiliar with Yoffa's technique (which I was before reading your paper, so thank you for the opportunity to improve my knowledge on the topic), then it is unclear that your study is actually comparing the supra and infra clavicular approach to subclavian vein access alone. This could be clarified with a short section describing the two techniques you are comparing clearly. The Plos-one readership is generalist.

Methods:

1. The time period for collection is short - By your results, the volume of cases in your centre is high. The numbers you have included exceed those in other literature. However your analysis would be even more powerful with greater numbers.

2. The study is not randomised. This is a major limitation, and could be resolved easily. I base this on the methodology stating the team used one technique first for a month, then simply switched over. Why not randomised?

3. The follow up for secondary outcomes is over a year earlier than the study period. Is this a typo?

"Secondary outcomes were surgery related complications either categorized as early

complications, defined by the occurrence of infection, a hematoma or pneumothorax

in the first week following the procedure and prior to the first use of the device, or late

complication after assessment in march 2016 such as infection,"

Results:

You start by using the term subclavicular, whereas you've used infraclavicular throughout the text.

Discussion:

This statement is not accurate - reference 17 specifically identifies benefits of US guidance for subclavian access from a recent meta-analysis, although not significant.

"US guidance is associated with frequent

risk of posterior vessel wall penetration as well as more lateral puncture sites as in the

landmark puncture technique which may lead to pleural injury."

Minor comments:

1. Abstract:

ICA is used without prior reference to the phrase it represents

The complications being studied are the primary outcome (I assume?). Please include information in the abstract.

Author needs to confirm - was the jugular approach also performed without US?

Some minor spelling mistakes

6. PLOS authors have the option to publish the peer review history of their article (what does this mean?). If published, this will include your full peer review and any attached files.

Reviewer #1: No

Reviewer #2: **Yes: **RA Benson

---

## [Author Response · Author response to Decision Letter 0]

18 Sep 2020

Dear editors

We would like to thank you for your feedback and inform you that all comments were followed.

Firstly, we would like to inform you that we are submitting our minimal underlying data set as an additional supplementary file according to PLOS journals requirement. All potentially identifiable patient information was fully anonymized.

Secondly, the responses to reviewer comments are the following: 

Reviewer #1:

1- Please mention the type of the study method in the title.

The required change has been made. 

2- Introduction: The hypothesis and objective of the study should be explained in detail in the introduction part.

The required change has been made. 

3- Methods: The sample size may not be sufficient to determine rarely seen early and late complications.

Indeed the sample size is a limitation of our study which was mentioned in our limitations paragraph in the discussion. However, a future study with a bigger sample of patients should take place. 

4- In the sentence: “Group A patients received TIVAD through a supraclavicular approach and group B through an infraclavicular approach in the first and second half of our study period respectively (see details below).” Which details below where? A table? Insertion techniques?

The mention refers to the Table 1 in the results section and was rectified. 

5- Results : Duration of catheter placement was mentioned in discussion part but there is no data in the results. 

The duration of catheter placement was already mentioned in paragraph 2 from the results section as well as the results section, Table 2, under the Mean operative time. 

6- Discussion : Personally, I do not agree with the authors about the presumption that ultrasound guided catheter placement may lead to deskilling. Although this is arguable, I believe that us guided procedures increase anatomically perception of the performers regarding the anatomic structures as they see them every time they perform the procedure (see anatomic variations as well).

This section was modified. 

Reviewer #2: 

1- Abstract: "yet the supraclavicular technique has been underused and never compared to the other method". No true - you yourselves reference several published comparisons!

indeed the different approaches in central line placement have been previously discussed, however, this is the first comparison of both techniques in port catheter implantation. 

2- Abstract: (1,4% vs 7% p=0,21). This is a large difference without significance. Ranges should be included with the data to understand why this is so. In the results table the figure is 1.4 vs 1.5. Typo?

This was a typo which we rectified. 

3- The text needs clarity. The techniques you are talking about need to be made very clear in the introduction, as at the moment it is a little confusing for me as a reader. For example: Be clear in the abstract and the introduction that you are just assessing access to the subclavian vein.

"Accordingly, the most frequent central venous catheterization approaches vary between the subclavian which is often adopted during surgical procedures and the internal jugular used for resuscitation. The use of the supraclavicular approach on the other hand is less usual. In fact, the supraclavicular technique for central vein puncture was first described In 1965 by Yoffa [2]."

The phrasing of this paragraph makes it seem like you are discussing subclavian access to supraclavicular IJV access, due to structure. If unfamiliar with Yoffa's technique (which I was before reading your paper, so thank you for the opportunity to improve my knowledge on the topic), then it is unclear that your study is actually comparing the supra and infra clavicular approach to subclavian vein access alone. This could be clarified with a short section describing the two techniques you are comparing clearly. The Plos-one readership is generalist.

The required change has been made to the abstract and introduction. 

4- Methods: The time period for collection is short - By your results, the volume of cases in your centre is high. The numbers you have included exceed those in other literature. However your analysis would be even more powerful with greater numbers.

We agree that a bigger sample would benefit and support this technique better. This retrospective study aimed to present a preliminary scientific basis for future randomized trials since no previous paper describes this approach for catheter implantation.

5- Methods: The study is not randomised. This is a major limitation, and could be resolved easily. I base this on the methodology stating the team used one technique first for a month, then simply switched over. Why not randomised?

We agree with your opinion, however, it is difficult to receive ethical approval for a randomized trial on a technique which was never presented in a scientific publication. As such, we reported this series to present this approach for the first time in literature. 

6- Methods: . The follow up for secondary outcomes is over a year earlier than the study period. Is this a typo?

This was a typo which we rectified. 

7- Results: You start by using the term subclavicular, whereas you've used infraclavicular throughout the text.

The required change has been made and the same term used throughout the manuscript. 

8- Discussion: This statement is not accurate - reference 17 specifically identifies benefits of US guidance for subclavian access from a recent meta-analysis, although not significant. "US guidance is associated with frequent risk of posterior vessel wall penetration as well as more lateral puncture sites as in the landmark puncture technique which may lead to pleural injury."

This statement was removed from the manuscript. 

9- Minor comments : Abstract: ICA is used without prior reference to the phrase it represents ; 

We referenced the abbreviation as requested. 

10- The complications being studied are the primary outcome (I assume?). Please include information in the abstract. 

We specified our primary outcomes in the abstract as requested. 

11- Author needs to confirm - was the jugular approach also performed without US? 

We would like to confirm that the jugular approach was also performed without US

12- Some minor spelling mistakes

The manuscript grammar and spelling were revised.

---

## [Decision Letter · Decision Letter 1]

20 Oct 2020

PONE-D-20-14056R1

Supraclavicular versus infraclavicular approach in inserting totally implantable central venous access for cancer therapy : a comparative study on safety, efficacy and intra interventional pain perception

PLOS ONE

Dear Dr. Souadka,

Thank you for submitting your manuscript to PLOS ONE. After careful consideration, we feel that it has merit but does not fully meet PLOS ONE’s publication criteria as it currently stands. Therefore, we invite you to submit a revised version of the manuscript that addresses the points raised during the review process.

Please revise the Title to be concise and informative.

Please address the reviewers' concerns and revise accordingly.

We look forward to receiving your revised manuscript.

Kind regards,

Academic Editor

PLOS ONE

Reviewers' comments:

Reviewer's Responses to Questions

**Comments to the Author**

1. If the authors have adequately addressed your comments raised in a previous round of review and you feel that this manuscript is now acceptable for publication, you may indicate that here to bypass the “Comments to the Author” section, enter your conflict of interest statement in the “Confidential to Editor” section, and submit your "Accept" recommendation.

Reviewer #1: (No Response)

Reviewer #3: (No Response)

2. Is the manuscript technically sound, and do the data support the conclusions?

Reviewer #1: Partly

Reviewer #3: (No Response)

3. Has the statistical analysis been performed appropriately and rigorously? 

Reviewer #1: No

Reviewer #3: (No Response)

4. Have the authors made all data underlying the findings in their manuscript fully available?

Reviewer #1: Yes

Reviewer #3: (No Response)

5. Is the manuscript presented in an intelligible fashion and written in standard English?

Reviewer #1: Yes

Reviewer #3: (No Response)

6. Review Comments to the Author

Reviewer #1: The authors are appreciated for their effort however, the required revisions were insufficiently done. Non-randomized characteristics of the study is a flaw. Ultrasound guidance is a standard for central venous access which is considered mandatory for increasing procedural success and reducing complications. Larger sample size is needed for assessing rare early and late complications.

Reviewer #3: (No Response)

7. PLOS authors have the option to publish the peer review history of their article (what does this mean?). If published, this will include your full peer review and any attached files.

Reviewer #1: No

Reviewer #3: No

---

## [Author Response · Author response to Decision Letter 1]

27 Oct 2020

Dear editors,

Firstly, we would like to thank you for your feedback and constructive criticism. As requested, we provided a shorter title for our manuscript. 

As regards the reviewer comment : “ Reviewer #1: The authors are appreciated for their effort however, the required revisions were insufficiently done. Non-randomized characteristics of the study is a flaw. Ultrasound guidance is a standard for central venous access which is considered mandatory for increasing procedural success and reducing complications. Larger sample size is needed for assessing rare early and late complications “ 

As previously mentioned, we did acknowledge the limitation originating from the non randomized character of the study in the limitations section as well as specified the need to consider our results with caution : “Limitations of this study are the retrospective aspect, absence of randomization and the fact that the results of this study were directly dependent on the senior surgeon's experience and skill. Therefore, caution is required when applying our results”

 We also acknowledge the sample size limitation, and the need for a larger sample in our future investigations regarding this subject. In fact, we believe this study provides us with guidance as we added to our limitation section. “ An additional limitation is the relatively small number of patients included, however, this will allow us to decide the required sample size for future studies addressing this technique.” 

On the other hand, we agree that ultrasound guidance is the standard of use for central venous access as mentioned in our conclusion. However, we made modifications to our conclusion to further emphasize on ultrasound guidance prioritization. In fact, central line placement is a procedure conducted by radiologists, intensive care specialists and surgeons with the latter using landmark techniques more frequently. This is particularly more frequent in low and middle income contexts where the availability of ultrasounds devices is not always possible, due to resource restrictions. In this case and only in the absence of ultrasound guidance is the yoffa approach preferred to other techniques.

---

## [Decision Letter · Decision Letter 2]

9 Nov 2020

Supraclavicular versus infraclavicular approach in inserting totally implantable central venous access for cancer therapy : a comparative retrospective study

PONE-D-20-14056R2

Dear Dr. Souadka,

We’re pleased to inform you that your manuscript has been judged scientifically suitable for publication and will be formally accepted for publication once it meets all outstanding technical requirements.

Kind regards,

Academic Editor

PLOS ONE

Additional Editor Comments (optional):

Reviewers' comments:

Reviewer's Responses to Questions

**Comments to the Author**

1. If the authors have adequately addressed your comments raised in a previous round of review and you feel that this manuscript is now acceptable for publication, you may indicate that here to bypass the “Comments to the Author” section, enter your conflict of interest statement in the “Confidential to Editor” section, and submit your "Accept" recommendation.

Reviewer #4: All comments have been addressed

Reviewer #5: All comments have been addressed

2. Is the manuscript technically sound, and do the data support the conclusions?

Reviewer #4: Partly

Reviewer #5: Yes

3. Has the statistical analysis been performed appropriately and rigorously? 

Reviewer #4: Yes

Reviewer #5: I Don't Know

4. Have the authors made all data underlying the findings in their manuscript fully available?

Reviewer #4: Yes

Reviewer #5: Yes

5. Is the manuscript presented in an intelligible fashion and written in standard English?

Reviewer #4: Yes

Reviewer #5: Yes

6. Review Comments to the Author

Reviewer #4: The paper is well revised. It deserves a wide scale reders exposure. Although the sample size is small but still it provides valuble information about TIVAP.

Reviewer #5: I read the article of the authors titled "Supraclavicular versus infraclavicular approach in inserting totally implantable central venous access for cancer therapy : a comparative retrospective study" with interest. When I examined the answers given to the previous criticisms, I saw that the necessary corrections were made.

7. PLOS authors have the option to publish the peer review history of their article (what does this mean?). If published, this will include your full peer review and any attached files.

Reviewer #4: **Yes: **Aram Baram

Reviewer #5: No

---

## [Editor Report · Acceptance letter]

11 Nov 2020

PONE-D-20-14056R2 

Supraclavicular versus infraclavicular approach in inserting totally implantable central venous access for cancer therapy : a comparative retrospective study 

Dear Dr. Souadka:

I'm pleased to inform you that your manuscript has been deemed suitable for publication in PLOS ONE. Congratulations! Your manuscript is now with our production department. 

Kind regards, 

on behalf of

Dr. Robert Jeenchen Chen 

Academic Editor

PLOS ONE